# SOAR: Improved Indexing for Approximate Nearest Neighbor Search

**Philip Sun, David Simcha, Dave Dopson, Ruiqi Guo, and Sanjiv Kumar**
Google Research
`{sunphil,dsimcha,ddopson,guorq,sanjivk}@google.com`

## Abstract

This paper introduces SOAR: **S**pilling with **O**rthogonality-**A**mplified **R**esiduals, a novel data indexing technique for approximate nearest neighbor (ANN) search. SOAR extends upon previous approaches to ANN search, such as spill trees, that utilize multiple redundant representations while partitioning the data to reduce the probability of missing a nearest neighbor during search. Rather than training and computing these redundant representations independently, however, SOAR uses an *orthogonality-amplified residual* loss, which optimizes each representation to compensate for cases where other representations perform poorly. This drastically improves the overall index quality, resulting in state-of-the-art ANN benchmark performance while maintaining fast indexing times and low memory consumption.

## 1 Introduction

The $k$-nearest neighbor search problem is defined as follows: we are given an $n$-item dataset $\mathcal{X} \in \mathbb{R}^{n \times d}$ composed of $d$-dimensional vectors, and for a query $q \in \mathbb{R}^d$, we would like to return the $k$ vectors in $\mathcal{X}$ closest to $q$. This problem naturally arises from a number of scenarios that require fast, online retrieval from vector databases; such applications include recommender systems [7], image search [14], and question answering [10], among many others.

While nearest neighbor search may be easily implemented with a linear scan over the elements of $\mathcal{X}$, many applications of nearest neighbor search utilize large datasets for which a brute-force approach is computationally intractable. The rapid development of deep learning and embedding techniques has been an especially strong driver for larger $k$-nearest neighbor datasets. For instance, multiple recent large language model (LLM) works incorporate external information by using $k$-nearest neighbors to retrieve from longer contexts [17] or large text corpora [5]. While datasets from typical applications a decade ago (e.g. Netflix) had sizes of around 1 million [15], the standard evaluation datasets from `big-ann-benchmarks` [16] all have 1 billion vectors with hundreds of dimensions.

These large datasets, in conjunction with the *curse of dimensionality*, a phenomenon which often makes it impossible to find the exact nearest neighbors without resorting to a linear scan, has led to a focus on approximate nearest neighbor (ANN) search, which can trade off a small search accuracy loss for a significant increase in search throughput.

A number of indexing schemes proposed for ANN search, including spill trees [12], assign datapoints to multiple portions of the index, such that the overall index provides a replicated, non-disjoint ("spilled") view of the dataset. These algorithms have used partitioning schemes amenable to random analysis, so that the replication factor can be provably shown to exponentially reduce the probability of missing a nearest neighbor. However, partitioning high-dimensional data is difficult, even without the constraint of doing so with a method amenable to randomized analysis; such constrained partitioning techniques have led to inferior quality indices that have not yielded good compute-accuracy tradeoffs.

37th Conference on Neural Information Processing Systems (NeurIPS 2023).

On the other hand, another family of approaches, those leveraging vector quantization (VQ), have demonstrated excellent empirical ANN performance, but little research has been done exploring the use of multiple randomized initializations of VQ indices to further increase ANN search efficiency. This is partly due to the fact that multiple VQ indices initialized through different random seeds do not have strong statistical guarantees and cannot be proven independent in their failure probabilities. This paper bridges the gap between these two approaches to ANN:

- We demonstrate the weaknesses of current VQ-based ANN indices and illustrate how multiple VQ indices may be used in conjunction to improve ANN search efficiency.

- We show that the naive training of multiple VQ indices for a single dataset leads to correlation in failure probabilities among the indices, limiting ANN search efficiency uplift, and present *spilling with orthogonality-amplified residuals* (SOAR) to ameliorate this correlation.

- We benchmark SOAR and achieve state-of-the-art performance, outperforming standard VQ indices, spilled VQ indices trained without SOAR, and all other approaches to ANN search that were submitted to the benchmark.

## 2   Preliminaries and Notation

### 2.1   Maximum inner product search (MIPS)

SOAR applies to the subclass of nearest neighbor search known as maximum inner product search, defined as follows for a query $q$ and a dataset $\mathcal{X}$:

$$\text{MIPS}_k(q, \mathcal{X}) = k\text{-}\underset{x \in \mathcal{X}}{\arg\max} \, \langle q, x \rangle.$$

Many nearest neighbor problems arise in the MIPS space naturally [6], and a number of conversions exist [4] from other commonly used ANN search metrics, such as Euclidean and cosine distance, to MIPS, and vice versa. We measure MIPS search accuracy using recall@$k$, defined as follows: if our algorithm returns the set $S$ of $k$ candidate nearest neighbors, its recall@$k$ equals $|\text{MIPS}_k(q, \mathcal{X}) \cap S|/k$. Additionally, we introduce the following notation to assist with MIPS analysis:

$$\text{RANK}(q, v, \mathcal{X}) = \sum_{x \in \mathcal{X}} \mathbb{1}_{\langle q, v \rangle \leq \langle q, x \rangle}.$$

The max inner product's RANK is 1; assuming no ties, $\text{RANK}(q, v, \mathcal{X}) \in [1, k]$ for $v \in \text{MIPS}_k(q, \mathcal{X})$.

### 2.2   Vector quantization (VQ)

Vector quantization can be leveraged to construct data structures that effectively prune the ANN search space; these data structures are commonly known as inverted file indices (IVF) or k-means trees. Vector-quantizing a dataset $\mathcal{X}$ produces two outputs:

- $\mathcal{C} \in \mathbb{R}^{c \times d}$, the codebook containing the $c$ partition centers.
- $\pi(v) : \mathbb{R}^d \mapsto \{1, \ldots, c\}$, the partition assignments that map each vector in $\mathcal{X}$ to one of the partition centers in $\mathcal{C}$. Oftentimes, $\pi(v)$ is defined as $\arg\min_{i \in \{1,\ldots,c\}} \|v - \mathcal{C}_i\|^2$, although other assignment functions may also be used.

We can then construct an inverted index over $\pi$; for each partition $i$, we store the set of datapoint indices belonging to that partition: $\{j | \pi(\mathcal{X}_j) = i\}$. Then, instead of computing $\text{MIPS}_k(q, \mathcal{X})$ directly, we may first compute $\text{MIPS}_{k'}(q, \mathcal{C})$, which is much faster because $|\mathcal{C}| \ll |\mathcal{X}|$. We can then use our inverted index data structure to further evaluate the datapoints within the top $k'$ partitions.

### 2.2.1   The k-means recall (KMR) curve

The effectiveness of the VQ-based pruning approach depends on the rank of the partitions that $\text{MIPS}_k(q, \mathcal{X})$ are in. If these partitions rank very well, we can set $k'$ very low, and search through few partitions while still achieving great MIPS recall. If these partitions rank poorly, however, the algorithm will have to spend lots of compute searching through many partitions in order to find the nearest neighbors. We may quantify this effectiveness using the *k-means recall* (KMR) curve (named

because VQ indices are commonly trained through k-means, although KMR can be computed for any VQ index, not just those trained via k-means), defined as follows:

$$\text{KMR}_k(t; \mathcal{Q}, \mathcal{X}, \mathcal{C}, \pi) = \frac{1}{k \cdot |\mathcal{Q}|} \sum_{q \in \mathcal{Q}} \sum_{v \in \text{MIPS}_k(q, \mathcal{X})} \mathbb{1}_{\text{RANK}(q, \mathcal{C}_{\pi(v)}, \mathcal{C}) \leq t} \tag{1}$$

The KMR curve for a given query sample $\mathcal{Q}$, dataset $\mathcal{X}$, and VQ index $(\mathcal{C}, \pi)$ quantifies the proportion of MIPS nearest neighbors present in the top $t$ VQ partitions, over varying $t$. The KMR curve is non-decreasing, with $\text{KMR}_k(0) = 0$ and $\text{KMR}_k(|\mathcal{C}|) = 1$. A KMR curve that more quickly approaches 1 indicates superior index quality.

## 3 Method

The key insight of our work is the use of a novel loss function to assign a datapoint $x$ to multiple VQ partitions, such that these additional partitions effectively recover $x$ in the pathological cases when $x$ is a nearest neighbor for a query that $x$'s original VQ partition handles poorly. These partitions that include $x$ work together synergistically to provide greater ANN search efficiency than any single partition could alone. Below, we describe the motivations behind our new loss and this multiple assignment. The supplementary materials contains the source code to generate this section's plots.

### 3.1 Search difficulty and quantized score error

For a datapoint $x$, define $r$ as the *partitioning residual*, equal to $x - \mathcal{C}_{\pi(x)}$. The partition center $\mathcal{C}_{\pi(x)}$ is the quantized form of $x$, so the difference between the exact and the quantized inner product scores is

$$\langle q, x \rangle - \langle q, \mathcal{C}_{\pi(x)} \rangle = \langle q, x - \mathcal{C}_{\pi(x)} \rangle = \langle q, r \rangle,$$

which we denote the *quantized score error*. Consider the case when $x$ is a nearest neighbor for the query: $x \in \text{MIPS}_k(q, \mathcal{X})$. In this scenario, we would like $\text{RANK}(q, \mathcal{C}_{\pi(x)}, \mathcal{C})$ to be low, so that the algorithm may search just the top few partitions and find the nearest neighbor $x$. A high $\text{RANK}$ would lead to increased search difficulty, because the algorithm would have to search more partitions and therefore do more work to find $x$.

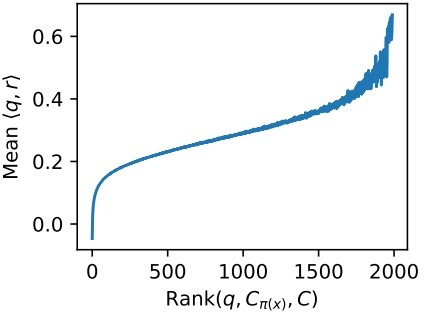

Figure 1: Greater search difficulty, as quantified by a higher $\text{RANK}(q, \mathcal{C}_{\pi(x)}, \mathcal{C})$, is associated with highly positive $\langle q, r \rangle$.

Given that $\langle q, \mathcal{C}_{\pi(x)} \rangle = \langle q, x \rangle - \langle q, r \rangle$ and that, by definition, $\langle q, x \rangle$ is large when $x \in \text{MIPS}_k(q, \mathcal{X})$, we know that $\langle q, \mathcal{C}_{\pi(x)} \rangle$ will be small (leading to greater search difficulty) when $\langle q, r \rangle$ is highly positive. Indeed, this can be confirmed empirically; in Figure 1 we plot the mean $\langle q, r \rangle$ as a function of $\text{RANK}(q, \mathcal{C}_{\pi(x)}, \mathcal{C})$ for all query-neighbor pairs $(q, x)$ in the Glove-1M dataset. The more difficult-to-find pairs have, on average, notably higher $\langle q, r \rangle$.

SOAR increases search efficiency in these situations when $x \in \text{MIPS}_k(q, \mathcal{X})$ and $\langle q, r \rangle$ is high.

### 3.2 Quantized score error decomposition

By the definition of the inner product,

$$\langle q, r \rangle = \|q\| \cdot \|r\| \cdot \cos \theta$$

where $\theta$ is the angle formed between the query and the partitioning residual. Without loss of generality, we may assume that $\|q\| = 1$ without any effect on the ranking of MIPS nearest neighbors. The two contributors to a highly positive $\langle q, r \rangle$ are therefore a large $\|r\|$ and $\cos \theta$ being near unity; reducing $\langle q, r \rangle$ requires targeting either, or both, of these contributors.

SOAR targets $\cos \theta$, because that term is both easier to reduce and has greater impact on $\langle q, r \rangle$:

- The VQ training loss already aims to minimize $\mathbb{E}[\|x - \mathcal{C}_{\pi(x)}\|^2] = \mathbb{E}[\|r\|^2]$, so further reductions in $\|r\|$ are difficult. In contrast, $\cos \theta$ is not directly optimized for, making it more amenable to reduction.

- $\cos\theta$ is typically much more strongly correlated with $\langle q, r \rangle$ than $\|r\|$ is, so its reduction has greater impact on search difficulty. Qualitatively, this stronger correlation can be explained by the far greater range of values $\cos\theta$ may take on, compared to $\|r\|$. The latter's values are concentrated between 0 and $\|x\|$, with not a very large ratio between the largest and smallest value. In comparison, $\cos\theta$ can take on any value in the range $[-1, 1]$; only this large multiplicative dynamic range can explain the variance in $\langle q, r \rangle$. Empirically, this can be seen demonstrated on Glove-1M in Figure 2 below.

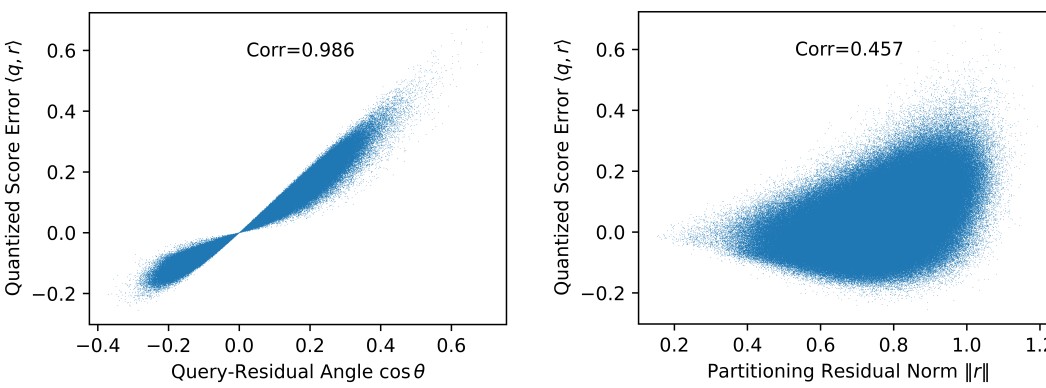

Figure 2: The cosine of the query-residual angle, $\cos\theta$ (**left**), is far more correlated with $\langle q, r \rangle$ than the residual norm $\|r\|$ (**right**), making the former a more promising target for reducing $\langle q, r \rangle$.

### 3.3 Spilled VQ assignment

We may attempt to mitigate the increase in search difficulty presented by high $\cos\theta$ by assigning each datapoint $x$ to a second VQ partition $\pi'(x)$, resulting in a second partitioning residual $r'$ that forms an angle $\theta'$ with the query. We denote this second assignment a *spilled assignment*.

This gives our ANN algorithm a "second chance"; assuming $\theta$ and $\theta'$ are independently distributed, the situations where $\cos\theta$ is high, thereby leading to high $\langle q, r \rangle$ and greater search difficulty, are unlikely to also have a high $\cos\theta'$. A low $\cos\theta'$ should lead to a low $\langle q, r' \rangle$ and should allow the ANN algorithm to find $x$ fairly efficiently. This idea can be extended to further (>2) assignments as well.

This approach leads to some implementation intricacies, as discussed in Section 3.5, but for now we focus on problems concerning theory when the spilled assignment $\pi'(x)$ is chosen so as to minimize $\|r'\|^2$.

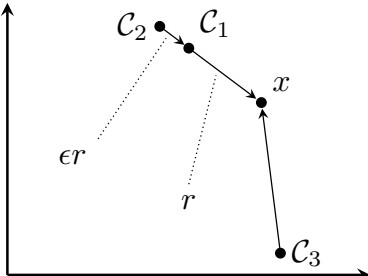

Figure 3: Naive spilled VQ assignment may be ineffective; selecting the two closest centroids $\mathcal{C}_1$ and $\mathcal{C}_2$ provides no benefit over using just $\mathcal{C}_1$.

This proposed approach fails when there is correlation between $\cos\theta$ and $\cos\theta'$, shown to the extreme in Figure 3. In this figure, if $\pi(x) = 1$, we may pick $\pi'(x) = 2$ because $\mathcal{C}_2$ is the second-closest centroid to $x$. However, $\mathcal{C}_2$ is collinear with $\mathcal{C}_1$ and $x$, resulting in $\theta = \theta'$. This leads to $\mathcal{C}_2$ acting as a strictly worse version of $\mathcal{C}_1$ for quantizing $x$: $\langle q, r' \rangle = (1 + \epsilon)\langle q, r \rangle$. The choice $\pi'(x) = 3$ would've been more effective, despite $\|x - \mathcal{C}_3\| > \|x - \mathcal{C}_2\|$, due to $\mathcal{C}_3$'s position giving a $\theta' \neq \theta$.

Experiments show that more than a theoretical concern, this is also a real-world issue. On the Glove-1M dataset, if we choose $\pi(x)$ and $\pi'(x)$ to be the closest and second-closest centroids by Euclidean distance, respectively, to $x$, we find a noticeable correlation between $\cos\theta$ and $\cos\theta'$, shown in Figure 4a. An even stronger correlation occurs between two VQ indices trained separately with different random seeds (Figure 4b). This correlation diminishes the benefit of spilled assignment, and reducing this correlation is critical to improving ANN search performance.

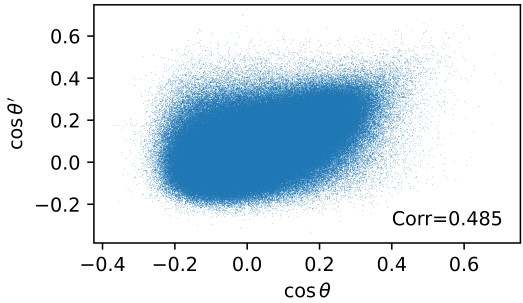 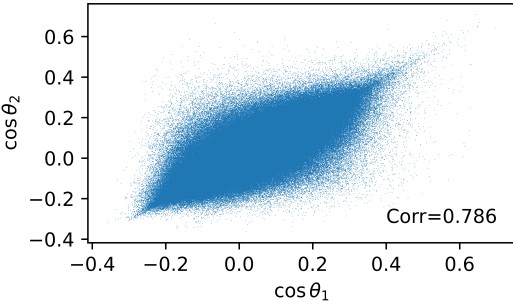

(a) $\theta$ and $\theta'$ come from a single VQ index, and are the closest and second-closest centroids, respectively.

(b) $\theta_1$ and $\theta_2$ come from two VQ indices, trained with different random seeds.

Figure 4: On Glove-1M, both a naive top-2 spilled assignment and two separately trained VQ indices exhibit noticeable correlation in query-residual angles; this reduces spilled assignment efficacy.

### 3.4 Spilling with orthogonality-amplified residuals

We now derive *spilling with orthogonality-amplified residuals* (SOAR), a novel VQ assignment loss that directly confronts the problem of correlated query-residual angles. To derive this loss, assume that the VQ centroids $\mathcal{C}$ and assignments $\pi$ are fixed. The goal of further spilled assignments $\pi'$ is to achieve low quantized score error $\langle q, r' \rangle$ specifically when the original spilled assignment has high $\cos\theta$, leading to high $\langle q, r \rangle$. In cases where $\cos\theta$ is low, the original assignments already achieve low quantized score error, so in such cases, it is less important to also have low $\langle q, r' \rangle$.

We now modify our assignment loss function to reflect this emphasis; rather than selecting $c' \in \mathcal{C}$ to minimize $\mathbb{E}_{q \in \mathcal{Q}}[(\langle q, x \rangle - \langle q, c' \rangle)^2] = \mathbb{E}_{q \in \mathcal{Q}}[\langle q, r' \rangle^2]$, we now also add a weighting term for $\cos\theta$:

$$\mathcal{L}(r', r, \mathcal{Q}) = \mathbb{E}_{q \in \mathcal{Q}}\left[w\left(\frac{\langle q, r \rangle}{\|r\| \cdot \|q\|}\right)\langle q, r' \rangle^2\right] = \mathbb{E}_{q \in \mathcal{Q}}[w(\cos\theta)\langle q, r' \rangle^2], \qquad (2)$$

where the weight function $w(\cdot) : \mathbb{R} \mapsto \mathbb{R}^{\geq 0}$ should be chosen to give greater emphasis to higher $\cos\theta$. To utilize this loss function in VQ assignment, we must evaluate the expectation, which leads to the following result:

**Theorem 3.1.** *For the weight function $w(t) = |t|^\lambda$ and a query distribution $\mathcal{Q}$ that is uniformly distributed over the unit hypersphere,*

$$\mathcal{L}(r', r, \mathcal{Q}) \propto \|r'\|^2 + \lambda\|\text{proj}_r r'\|^2.$$

*Proof.* See Appendix A.1. $\qquad\qquad\square$

By leveraging Theorem 3.1, we may efficiently perform spilled VQ assignment using our new loss by computing the squared $\ell_2$ distance to each centroid, adding a penalty term for parallelism between the original residual $r$ and the candidate residual $r'$, and taking the $\arg\min$ among all centroids. We present the following statements to help develop intuition about the SOAR loss:

**Corollary 3.1.1.** *For the uniform weight function $w(t) = 1 = |t|^0$, the SOAR loss is equivalent to standard Euclidean assignment: $\mathcal{L}(r', r, \mathcal{Q}) \propto \|r'\|^2$.*

The parameter $\lambda$ controls how much the spilled assignment $\pi'$ should prioritize quantization performance when the original assignment performs poorly, relative to general quantization performance. As $\lambda$ is increased, the former is further emphasized; at $\lambda = 0$, only the latter is considered, leading to standard Euclidean assignment. See Figure 9 in Experiments for a visualization of this tradeoff.

**Corollary 3.1.2.** *For a fixed $\|r'\|$, $\mathcal{L}$ is minimized when $r$ and $r'$ are orthogonal, in which case $\lambda\|\text{proj}_r r'\|^2 = 0$ and $\mathcal{L}(r', r, \mathcal{Q}) \propto \|r'\|^2$.*

Our technique's name originates from the observation that residuals are encouraged to be orthogonal.

**Lemma 3.2.** $\|\text{proj}_r r'\| = \|r'\| \cdot \rho_{\langle q, r \rangle, \langle q, r' \rangle}$, *where $\rho$ is the Pearson correlation coefficient, and the correlation is computed over $q$ uniformly distributed over the hypersphere.*

The above lemma shows that in addition to the weighted quantization error derivation in Equation 2, we may also interpret the SOAR loss as adding a penalization term for correlation in quantized score error, scaled by the magnitude of the quantization error.

*Proof.*

$$\rho_{\langle q,r\rangle,\langle q,r'\rangle} = \frac{\mathrm{Cov}[\langle q,r\rangle, \langle q,r'\rangle]}{\sigma_{\langle q,r\rangle}\sigma_{\langle q,r'\rangle}} = \frac{\mathbb{E}[\langle q,r\rangle\langle q,r'\rangle] - \mathbb{E}[\langle q,r\rangle]\mathbb{E}[\langle q,r'\rangle]}{\sigma_{\langle q,r\rangle}\sigma_{\langle q,r'\rangle}},$$

and by symmetry over the hypersphere, $\mathbb{E}[\langle q,r\rangle] = \mathbb{E}[\langle q,r'\rangle] = 0$ so our expression simplifies to

$$\rho_{\langle q,r\rangle,\langle q,r'\rangle} = \frac{\mathbb{E}[\langle q,r\rangle\langle q,r'\rangle]}{\sigma_{\langle q,r\rangle}\sigma_{\langle q,r'\rangle}} = \frac{\mathbb{E}[r^T qq^T r']}{\sigma_{\langle q,r\rangle}\sigma_{\langle q,r'\rangle}} = \frac{r^T\mathbb{E}[qq^T]r'}{\sigma_{\langle q,r\rangle}\sigma_{\langle q,r'\rangle}}.$$

It's well known that for uniformly distributed, unit-norm $q \in \mathbb{R}^d$ that $\mathbb{E}[qq^T] = I_d/d$, where $I_d$ is the $d$-dimensional identity matrix, and that $\sigma_{\langle q,v\rangle} = \|v\|/\sqrt{d}$ for any fixed $v$. We leverage this to get

$$\rho_{\langle q,r\rangle,\langle q,r'\rangle} = \frac{r^T(I_d/d)r'}{(\|r\|/\sqrt{d})(\|r'\|/\sqrt{d})} = \frac{r^T r'}{\|r\|\cdot\|r'\|} = \left\langle \frac{r}{\|r\|}, \frac{r'}{\|r'\|} \right\rangle = \frac{1}{\|r'\|}\|\mathrm{proj}_r r'\|.$$

$\square$

### 3.5 Implementation considerations

SOAR comes with some memory overhead relative to a non-spilled VQ MIPS index, due to its multiple assignments causing some duplication of data. The overhead is quite negligible because only the product-quantized (PQ) [9] datapoint representation, not the datapoint's highest-bitrate representation (usually int8 or float32), is duplicated. Since PQ is much more compressed than int8 or float32, duplicating the PQ data for each of a datapoint's spilled assignments only marginally increases memory footprint, as illustrated in Figure 5. This can be quantified analytically:

- SOAR increases memory consumption by $4 + \frac{d}{2s}$ bytes/datapoint, assuming 16 centers per subspace (usually chosen for amenability to SIMD) and $s$ dimensions per PQ subspace.

- The datapoint's original assignment already contained a PQ representation occupying $4 + \frac{d}{2s}$ bytes, and its highest-bitrate representation requires $d$ bytes (int8) or $4d$ bytes (float32).

When the ANN index's highest-bitrate datapoint representation is int8 or float32, the relative index size growth is therefore $\frac{4 + \frac{d}{2s}}{d + 4 + \frac{d}{2s}} \approx \frac{\frac{d}{2s}}{d + \frac{d}{2s}} = \frac{1}{2s+1}$ or $\frac{4 + \frac{d}{2s}}{4d + 4 + \frac{d}{2s}} \approx \frac{\frac{d}{2s}}{4d + \frac{d}{2s}} = \frac{1}{8s+1}$, respectively. Table 1 empirically corroborates these estimates and shows the memory cost is indeed quite low (5%-20% relative increase over a non-spilled VQ index).

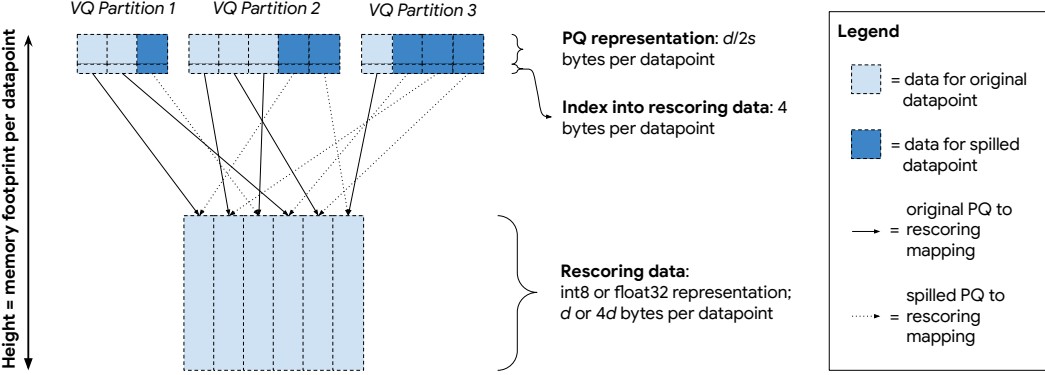

Figure 5: Memory layout changes for a SOAR-enabled ANN index. Memory footprint is proportional to area of colored cell. VQ centroid data (not shown) remains unchanged. We can see that the additional memory occupied by SOAR (dark blue) is low relative to the total memory consumption.

SOAR also results in a small amount of query-time CPU overhead over a non-spilled VQ index, due to SOAR's need for deduplication: a single datapoint may now appear in multiple partitions that are all searched. This overhead is low, though, because VQ inverted index performance is primarily bottlenecked by memory accesses to the VQ partition data, and this access pattern remains the same.

Finally, SOAR requires a minor tweak to the indexing pipeline. Creating a SOAR-enabled index first requires training a standard, non-spilled VQ index as usual. This gets used to populate the primary assignments, but is also used to calculate the residual $r$, which is then used in the SOAR loss to select further, spilled assignments. Other parts of the ANN indexing pipeline remain unchanged.

Overall, SOAR's search efficiency gains greatly outweigh its memory, CPU, and indexing overheads.

### 3.5.1 Spilling to further centroids

The analysis conducted in Section 3.4 focused specifically on the case of assigning to two partitions, $\pi$ and $\pi'$. This can be generalized to an arbitrary number of further assignments, where each subsequent assignment is done with a loss considering the distribution of quantized score errors from all prior assignments. We forgo this generalization in experiments and focus on the two-assignment setup because further assignments have diminishing returns; the first spilled assignment is generally sufficient to handle the cases where the primary assignment has high quantized score error, so further assignments are only needed in rare, doubly pathological situations. Meanwhile, the additional memory and indexing cost increases linearly with further assignments; two assignments strikes a good balance between these costs and KMR benefits while also keeping implementation simple.

## 4 Related Works

### 4.1 Spill trees

Spill trees (or sp-trees) were originally introduced in [12], and also assign datapoints to multiple vertices at the same level of the tree, just as SOAR may assign a datapoint to multiple VQ partitions. However, there are significant differences; spill trees were searched using *defeatist search*, a greedy, no-backtracking strategy that took the most promising root-to-leaf path. SOAR indices utilize backtracking in their search procedure; multiple partitions, not just the top one, are evaluated further.

Spill trees also differ in how they perform assignment; their multiple assignments occur at each level of the tree, such that a single datapoint could appear in exponentially many leaves, with respect to the tree depth (which was significant, as spill trees are binary and therefore rather deep). This led to a large (sometimes >100x [11]) storage overhead, which was very costly. In contrast, SOAR performs the multiple assignments at individual levels of the tree, leading to a storage overhead that is constant with respect to the tree depth, and much lower (typically 10%-20%; see Table 1). Finally, SOAR leverages its custom, spilling-aware assignment loss to decide which among the many VQ partitions to spill to, while the spill tree, as a consequence of its binary structure, only had to make the binary decision of whether to spill at all.

### 4.2 Graph-based algorithms

Graph-based approaches to ANN search, such as [13], have received significant amounts of recent research attention, and a number of such algorithms are featured in comparisons in Section 5.4. If one considers datapoints and partition centers to be vertices, and datapoint-to-partition assignments to be edges, a traditional VQ index would form a tree structure, while spilled VQ assignment (including SOAR) would transform the tree into a cycle-containing general graph data structure. In this sense, SOAR makes VQ indices more closely resemble graph-based ANN indices.

However, graph-based ANN algorithms (as defined in the traditional sense) have no concept of partition centers, and only add edges between datapoints themselves, leading to a lack of hierarchy and linearizability. SOAR inherits the linearizability of tree-based approaches, leading to predictable and sequential memory access patterns that are necessary for achieving maximal performance on modern hardware. SOAR also inherits the small index sizes more easily achievable with tree-based methods. When viewed as a graph, a SOAR index only uses two edges (two partition assignments) per datapoint, while traditional graph-based indices typically use dozens of edges per datapoint, which incurs significant storage overhead that SOAR avoids.

# 5 Experiments

## 5.1 KMR curve comparison

The KMR curve is a deterministic and easily computable metric for VQ-style ANN indices that's highly predictive of real-world performance. Below, we plot $\text{KMR}_k(t)$ as defined in Equation 1, but rather than plotting with respect to $t$, we plot with respect to the sum of the sizes of the $t$ top-ranked partitions. This partition size weighting is necessary because spilled VQ data structures have more points per partition due to their multiple assignments, making each individual partition more expensive to search. The results are shown in Figure 6; further details are given in Appendix A.2.

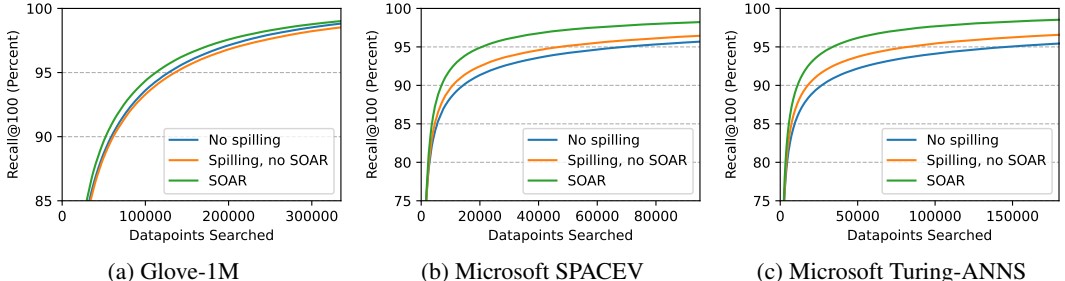

(a) Glove-1M        (b) Microsoft SPACEV        (c) Microsoft Turing-ANNS

Figure 6: SOAR allows a VQ index to achieve a given recall target while reading fewer total datapoints and therefore utilizing less memory bandwidth, increasing ANN search performance. SOAR's improvements are especially large on Microsoft SPACEV and Microsoft Turing-ANNS, the billion-scale datasets in this experiment; see Section 5.3 for further analysis of dataset size.

## 5.2 Correlation analysis

Here, we look at a number of statistics relating to the quantized score error that shed light on how SOAR improves KMR and overall VQ index quality. These statistics were computed on the Glove-1M dataset with the SOAR $\lambda$ set to 1; see source code in supplementary materials for more details.

Figure 7 looks at the query-residual angle, where the residuals are taken from the nearest neighbor results for each query. The scatterplot looks at $\cos\theta$ from the primary VQ assignment and $\cos\theta'$ from the spilled VQ done with SOAR. In comparison with the equivalent scatterplots in Figure 4 done without SOAR, we can see that SOAR significantly reduces the correlation between the cosines of the two angles, thus increasing the efficacy of multiple VQ assignments.

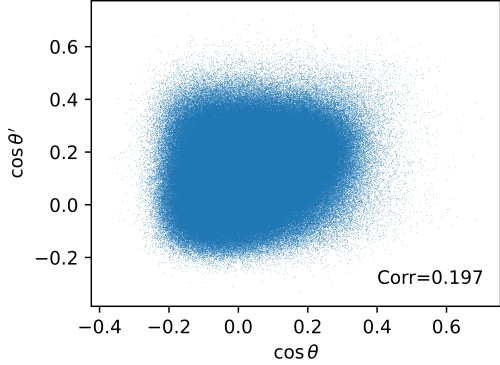

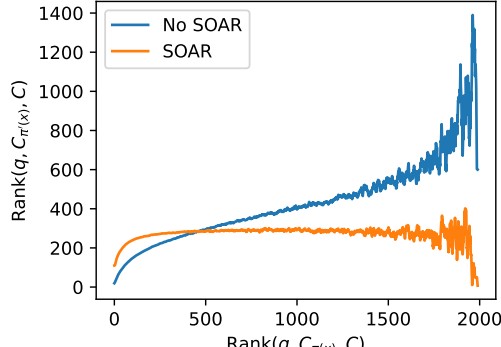

Figure 7: Scatter plot, analogous to Figure 4a, but with spilled assignments performed using SOAR loss; angular correlation is much lower.

Figure 8: The nearest neighbors most difficult to find under partitioning $\pi$, as quantified by $\text{RANK}(q, \mathcal{C}_{\pi(x)}, \mathcal{C})$, also tend to be difficult to find under $\pi'$ when SOAR isn't used.

The end result of this reduced correlation can be seen in Figure 8, which plots the mean rank of a nearest neighbor's spilled centroid against the rank of that neighbor's primary centroid. We can

see that without SOAR, when a nearest neighbor's primary partition $\pi(x)$ ranks poorly, the nearest neighbor's spilled partition $\pi'(x)$ tends to also rank poorly, meaning the spilled assignment is doing little to help ANN performance; the ANN algorithm will still have to search through many partitions to find $x$. With SOAR, $\text{RANK}(q, \mathcal{C}_{\pi'(x)}, \mathcal{C})$ remains low, allowing the ANN algorithm to search significantly fewer partitions to find the nearest neighbors.

Next, in Figure 9, we investigate the effect of SOAR's $\lambda$ parameter on the resulting VQ index. We can see that as $\lambda$ is increased, the resulting partitioning $\pi'$ from SOAR will have quantized score error $\langle q, r' \rangle$ less and less correlated with the quantized score error $\langle q, r \rangle$ from $\pi$. This is beneficial for ANN search efficiency, but on the other hand, $\mathbb{E}\left[\left\| x - \mathcal{C}_{\pi'(x)} \right\|^2 \right] = \mathbb{E}\left[\left\| r' \right\|^2 \right]$ increases with respect to $\lambda$. The diminishing relative importance of $\left\| r' \right\|^2$ as $\lambda$ increases is a natural consequence of Theorem 3.1, but nonetheless results in a general increase in magnitude of $\langle q, r' \rangle$ and is detrimental to ANN search accuracy; setting the optimal $\lambda$ requires balancing these effects.

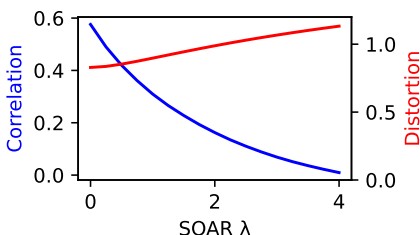

Figure 9: Raising SOAR $\lambda$ increases VQ distortion $\mathbb{E}[\|r'\|^2]$, but lowers score correlation $\rho_{\langle q,r \rangle, \langle q,r' \rangle}$.

### 5.3 Effects of dataset size and recall target

In Figure 10, we take various size samples from the `ann-benchmarks.com` DEEP dataset and compare the ratio of how many datapoints a SOAR VQ index must access compared to a standard VQ index to achieve the same recall. This ratio is the dependent variable in Figure 10; a higher ratio indicates a greater improvement from SOAR. We can see that higher recall targets, and larger samples, lead to greater ratios.

We maintain a ratio of 400 datapoints per partition across all sample sizes (for example, the 1M datapoint sample was trained with 2500 partitions). This sampling-based approach was chosen over using various datasets of differing sizes, to eliminate effects due to varying search difficulties among datasets.

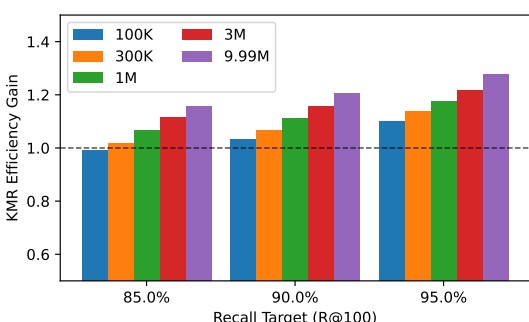

Figure 10: SOAR's ANN search efficiency benefit grows as SOAR is applied to larger datasets, and as the ANN recall target rises.

### 5.4 End-to-end recall-speed benchmarks

ANN search algorithms are ultimately compared by their ability to trade off ANN search accuracy (measured by recall), and speed, measured by queries-per-second throughput. Here we perform this comparison between SOAR and state-of-the-art algorithms in two standardized benchmarking frameworks.

First, we modify ScaNN [8] to utilize SOAR and benchmark its performance on a standardized testing environment with an Intel Xeon W-2135 processor and the test setup from ann-benchmarks.com [3]. The algorithms for comparison were installed, configured, and tuned according to their respective benchmark submissions, and run on the same environment; see Appendix A.3 for more details. As shown in Figure 11, SOAR improves upon ScaNN's already state-of-the-art performance.

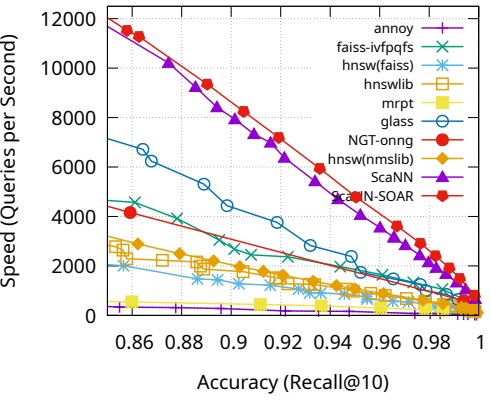

Figure 11: SOAR + ScaNN on Glove-1M.

Next, we benchmark an ANN search implementation using SOAR on a multi-level k-means clustering against submissions to Track 3 of big-ann-benchmarks.com [16]. These benchmarks allow the use of custom hardware, enabling greater flexibility in approaches to ANN search, but also complicating direct comparison due to the difficulty in accounting for varying hardware costs. In an effort to present existing benchmark submissions in a fair light, we compare SOAR against these submissions in two ways:

1. We compare the ratio of search throughput to hardware price. The ratios for competing algorithms were taken directly from the benchmark leaderboard, which was straightforward, but SOAR's results were from virtualized cloud compute, so the capital expenditure of buying a server with equivalent power had to be estimated. Hardware prices now may be slightly lower than prices used by other submissions in 2022, their time of submission; on the other hand, SOAR likely would've performed better if run on dedicated hardware, where no other jobs are competing for the processor's various shared resources.

2. We compare the ratio of search throughput to estimated monthly cloud billing cost for each submission's hardware. One benefit of this method is that hardware costs are completely equalized; in contrast, the alternate method of comparison rewarded contestants for finding especially discounted offerings for the same memory or compute. Additionally, the popularity of cloud infrastructure over self-hosting implies this ratio is what truly matters for many deployers of ANN search algorithms. However, we can't compare against submissions that use proprietary or deprecated hardware, and benchmark contestants never directly optimized for this ratio, because cloud cost was never used in any original benchmark leaderboard.

Although neither comparison alone is perfect, the two in aggregate provide a solid characterization of SOAR's performance. Further experimental details are provided in Appendix A.4, and results are shown in Figure 12 below; our index leads under both cost metrics, and SOAR is critical for our index's performance, roughly doubling throughput over a traditional, non-spilled VQ index.

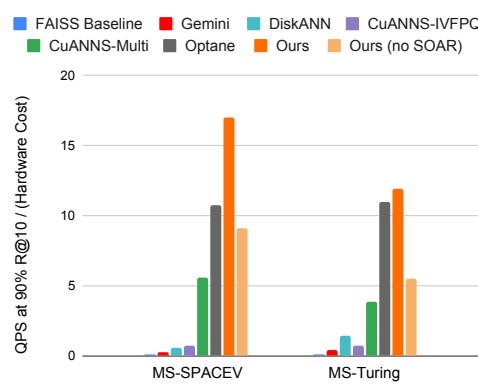 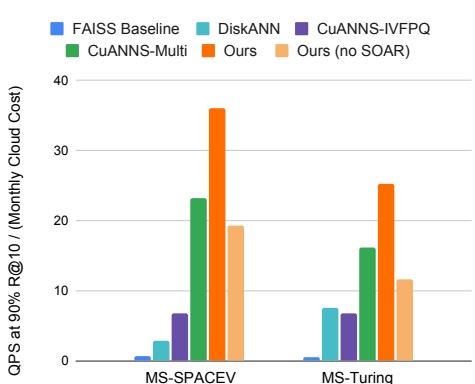

(a) Throughput-to-capex ratio (higher is better)  (b) Throughput-to-cloud bill ratio (higher is better)

Figure 12: SOAR leads big-ann-benchmarks performance under both methods of comparison.

Finally, we present memory consumption figures for the ANN indices in these benchmarks in Table 1. The minor increase in memory usage is discussed and analyzed in Section 3.5, and we find that analysis corroborates well with the empirical measurements below. For all three datasets, the index size increase is very minor, and the throughput gains from SOAR could allow the same query traffic to be served by fewer server replicas, in fact potentially *reducing* net memory consumption.

| Dataset | Memory Usage, No SOAR | Memory Usage with SOAR |
|---|---|---|
| Glove-1M | 453.5 MB | 488.4 MB (+7.7%) |
| Microsoft Turing-ANNS | 120.03 GB | 140.23 GB (+16.8%) |
| Microsoft SPACEV | 120.85 GB | 141.80 GB (+17.3%) |

Table 1: ANN index memory consumption before/after SOAR.

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

# A Appendix

## A.1 Proof of Theorem 3.1

We may expand our expectation as follows:

$$\mathcal{L}(r', r, \mathcal{Q}) = \mathbb{E}_{q \in \mathcal{Q}} \left[ w \left( \frac{\langle q, r \rangle}{\|r\| \cdot \|q\|} \right) \langle q, r' \rangle^2 \right]$$

$$= \int_0^\pi w(\cos\theta) \mathbb{E}_{q \in \mathcal{Q}} \left[ \langle q, r' \rangle^2 \mid \frac{\langle q, r \rangle}{\|r\| \cdot \|q\|} = \cos\theta \right] dP \left[ \frac{\langle q, r \rangle}{\|r\| \cdot \|q\|} < \cos\theta \right].$$

To evaluate this integral, first decompose $q$ and $r'$ into components parallel and orthogonal to $r$; let us call these components $q_\|$, $q_\perp$, $r'_\|$, and $r'_\perp$, respectively. This allows us to simplify the inner expectation:

$$\mathbb{E}_{q \in \mathcal{Q}} \left[ \langle q, r' \rangle^2 \mid \frac{\langle q, r \rangle}{\|r\| \cdot \|q\|} = \cos\theta \right] = \mathbb{E}_{q \in \mathcal{Q}} [\langle q, r' \rangle^2 \mid \|q_\|\| = \cos\theta]$$

$$= \mathbb{E}_{q \in \mathcal{Q}} \left[ \left\langle q_\| + q_\perp, r'_\| + r'_\perp \right\rangle^2 \mid \|q_\|\| = \cos\theta \right]$$

$$= \mathbb{E}_{q \in \mathcal{Q}} \left[ \left( \left\langle q_\|, r'_\| \right\rangle + \langle q_\perp, r'_\perp \rangle \right)^2 \mid \|q_\|\| = \cos\theta \right]$$

$$= \mathbb{E}_{q \in \mathcal{Q}} [(\cos\theta \|r'_\|\| + \langle q_\perp, r'_\perp \rangle)^2 \mid \|q_\|\| = \cos\theta]$$

$$= \cos^2\theta \|r'_\|\|^2 + 2\cos\theta \|r'_\|\| \left\langle \mathbb{E}[q_\perp \mid \|q_\|\| = \cos\theta], r'_\perp \right\rangle$$
$$+ \mathbb{E}[\langle q_\perp, r'_\perp \rangle)^2 \mid \|q_\|\| = \cos\theta]$$

$$= \cos^2\theta \|r'_\|\|^2 + \sin^2\theta \|r'_\perp\|^2/(d-1).$$

Meanwhile, $dP \left[ \frac{\langle q, r \rangle}{\|r\| \cdot \|q\|} < \cos\theta \right]$ is proportional to the surface area of a $(d-1)$-dimensional hypersphere of radius $\sin\theta$, which we may express as $A \sin^{d-2}\theta$ for some constant $A$. Our integral then becomes

$$\mathcal{L}(r', r, \mathcal{Q}) = \int_0^\pi w(\cos\theta) \left[ \cos^2\theta \|r'_\|\|^2 + \sin^2\theta \|r'_\perp\|^2/(d-1) \right] A \sin^{d-2}\theta d\theta$$

$$= \left( A \int_0^\pi w(\cos\theta) \sin^{d-2}\theta \cos^2\theta \right) \|r'_\|\|^2 +$$

$$\left( A \int_0^\pi w(\cos\theta) \sin^d\theta \right) \|r'_\perp\|^2/(d-1).$$

Now define $I_d = A \int_0^\pi w(\cos\theta) \sin^d\theta$; note that $\mathcal{L}(r', r, \mathcal{Q}) = (I_{d-2} - I_d) \|r'_\|\|^2 + I_d \|r'_\perp\|^2/(d-1)$. For the weight function $w(t) = |t|^\lambda$, we find that $I_d$ has a recursive definition if we utilize integration by parts with $v = \cos\theta$ and $u = -\cos^\lambda\theta \sin^{d-1}\theta$:

$$I_d = A \int_0^\pi |\cos\theta|^\lambda \sin^d\theta d\theta$$

$$= 2A \int_0^{\pi/2} \cos^\lambda\theta \sin^d\theta d\theta$$

$$= -2A \cos^{\lambda+1}\theta \sin^{d-1}\theta \Big|_0^{\pi/2}$$

$$\quad - 2A \int_0^{\pi/2} \cos\theta \Big( -(d-1)\cos^{\lambda+1}\theta \sin^{d-2}\theta - \lambda\sin^d\theta\cos^{\lambda-1}\theta \Big) d\theta$$

$$= 2A(d-1) \int_0^{\pi/2} \cos^{\lambda+2}\theta \sin^{d-2}\theta d\theta - 2\lambda A \int_0^{\pi/2} \cos^\lambda\theta \sin^d\theta d\theta$$

$$= 2A(d-1) \int_0^{\pi/2} \cos^\lambda(1 - \sin^2\theta)\theta \sin^{d-2}\theta d\theta - \lambda I_d$$

$$= (d-1)I_{d-2} - (d-1)I_d - \lambda I_d.$$

Combining terms, we have $(d+\lambda)I_d = (d-1)I_{d-2}$; this implies for our loss that

$$\mathcal{L}(r', r, \mathcal{Q}) = (I_{d-2} - I_d)\left\|r'_\parallel\right\|^2 + I_d\|r'_\perp\|^2/(d-1)$$

$$= \left(\frac{d+\lambda}{d-1} - 1\right) I_d \left\|r'_\parallel\right\|^2 + I_d\|r'_\perp\|^2/(d-1)$$

$$= \frac{I_d}{d-1} \cdot \left( (\lambda+1)\left\|r'_\parallel\right\|^2 + \|r'_\perp\|^2 \right)$$

$$= \frac{I_d}{d-1} \cdot \left( \lambda\left\|r'_\parallel\right\|^2 + \|r'\|^2 \right)$$

$$\propto \lambda\left\|r'_\parallel\right\|^2 + \|r'\|^2.$$

Note that $r'_\parallel = \text{proj}_r r'$, so this is our desired result. This is very similar to the analysis behind Theorem 3.3 of [8].

## A.2 KMR curve comparison: further details

The Glove-1M dataset came from ann-benchmarks.com [3], while the Microsoft SPACEV and Microsoft Turing-ANNS datasets came from big-ann-benchmarks.com.

- Glove-1M was trained on an anisotropic loss [8] with 2000 partitions, and SOAR was run with $\lambda = 1$.
- The two billion-datapoint datasets were trained on an anisotropic loss with approximately 7.2 million partitions, and SOAR was run with $\lambda = 1.5$.

Table 2 below presents the approximate number of datapoints needed to search in order to achieve various recall targets on these datasets, as well as SOAR's KMR gain over non-spilled VQ indices.

| Dataset | Recall Target (R@100) | No Spilling | Spilling, No SOAR | SOAR | KMR gain, SOAR over No Spilling |
|---|---|---|---|---|---|
| Glove-1M | 80% | 19983 | 21126 | 18352 | 1.09x |
| | 85% | 32639 | 34109 | 29369 | 1.11x |
| | 90% | 59063 | 61392 | 52292 | 1.13x |
| | 95% | 127819 | 135612 | 112488 | 1.14x |
| Microsoft SPACEV | 80% | 2850 | 2700 | 2450 | 1.16x |
| | 85% | 5550 | 4650 | 3700 | 1.50x |
| | 90% | 14350 | 10850 | 6950 | 2.06x |
| | 95% | 69050 | 47050 | 20550 | 3.36x |
| Microsoft Turing-ANNS | 80% | 4500 | 3950 | 3400 | 1.32x |
| | 85% | 9350 | 7250 | 5600 | 1.67x |
| | 90% | 26700 | 17800 | 11150 | 2.39x |
| | 95% | 145900 | 82100 | 33800 | 4.32x |

Table 2: KMR curve results in tabulated form; see Figure 6 for graphs of this data.

## A.3 `ann-benchmarks.com` Glove-1M benchmark details

The `ann-benchmarks.com` benchmark results came from a VQ-PQ index; the VQ index had 2000 partitions and used SOAR $\lambda = 1$. Both VQ and PQ were trained with anisotropic loss. The PQ quantization was configured with 16 subspaces and $s = 2$ dimensions per subspace, and the highest-bitrate representation of the datapoints was encoded in 32-bit floats. By the analysis from Section 3.5, we would therefore expect SOAR to increase index size by $1/17 \approx 5.9\%$, which is quite close to the empirical measurement in Table 1.

## A.4 End-to-end recall-speed benchmarks: further details

### A.4.1 General index setup

The `big-ann-benchmarks.com` results used a multilayer VQ index; the lower VQ index had approximately 7.2 million partitions, and these partition centers were vector-quantized again to 40000 partitions. PQ-quantized forms of the VQ residuals were used as intermediate scoring stages. The highest-bitrate form of the dataset stored by the index was an INT8-quantized representation. All VQ, PQ, and INT8 quantizations were trained with anisotropic loss.

### A.4.2 Estimated cost of SOAR benchmark hardware

The SOAR benchmark results came from running on a server using 32 vCPU (16 physical cores) on an Intel Cascade Lake generation processor with 150GB of memory. The Supermicro SYS-510P-M configured with:

- 1 x Intel® Xeon® Silver 4314 Processor 16-Core 2.40 GHz 24MB Cache (135W)
- 6 x 32GB DDR4 3200MHz ECC RDIMM Server Memory (2Rx8 - 16Gb)
- 1 x 1TB 3.5" MG04ACA 7200 RPM SATA3 6Gb/s 128M Cache 512N Hard Drive

should provide an upper bound for the cost of the SOAR benchmark setup, because the Supermicro server as configured has a newer generation and higher-clocked processor, and more memory, than the actual virtualized hardware used in benchmarking SOAR. At the time of writing, such a Supermicro server costs $2740.60. Combining this with the `big-ann-benchmarks.com` results gives us the following table, used to produce the plots in Figure 12a:

| Algorithm | Hardware Cost | MS-SPACEV Queries/Sec (90% R@10, from here) | MS-Turing Queries/Sec (90% R@10, from here) |
|---|---|---|---|
| FAISS Baseline | $22021.90 | 3265 | 2845 |
| DiskANN | $11742 | 6503 | 17201 |
| Gemini | $55726.66 | 16422 | 21780 |
| CuANNS-IVFPQ | $150000 | 108302 | 109745 |
| CuANNS-Multi | $150000 | 839749 | 584293 |
| OptANNe GraphANN | $14664.20 | 157828 | 161463 |
| Ours | $2740.60 | 46712 | 32608 |

### A.4.3 Cloud cost details

The monthly cloud infrastructure costs were derived from Google Compute Engine's on-demand pricing in the `us-central1` region. At the time of writing, the costs were [1]:

| Item | Monthly Cost (USD) |
|---|---|
| 1 vCPU | $24.81 |
| 1GB Memory | $3.33 |
| 1GB Local SSD | $0.08 |
| A100 80GB | $2868.90 |
| V100 16GB | $1267.28 |

Using this information, we computed the monthly cloud spend required for each of the entries to Track 3 of the `big-ann-benchmarks` competition:

| Algorithm | vCPU | RAM (GB) | Other | Total Monthly Cost (USD) |
|---|---|---|---|---|
| FAISS Baseline | 32 | 768 | 1x V100 16GB | $4617.57 |
| DiskANN | 72 | 64 | 3276.8GB SSD | $2261.18 |
| CuANNS-IVFPQ | 256 | 2048 | 1x A100 80GB[1] | $16036.46 |
| CuANNS-Multi | 256 | 2048 | 8x A100 80GB | $36118.76 |

Two submissions from the original `big-ann-benchmarks` competition could not be included in this comparison: Intel's OptANNe GraphANN submission, and GSI Technology's Gemini submission. The former relied on Intel Optane storage technology, which has been discontinued [2] and therefore isn't available on any mainstream cloud compute provider, and can't be priced. The latter leverages proprietary hardware not available to cloud providers. However, both of these entries are present in the throughput-per-capex ranking featured earlier.

Our own results came from a 32 vCPU, 150GB machine, which would cost $1293.09 per month. We achieved 32608 QPS on Microsoft Turing-ANNS and 46712 QPS on Microsoft SPACEV, leading to the numbers presented in Figure 12. These results are also reproduced in tabular form below.

| Algorithm | MS-SPACEV Queries/Sec (90% R@10, from here) | MS-SPACEV Throughput / Cost | MS-Turing Queries/Sec (90% R@10, from here) | MS-Turing Throughput / Cost |
|---|---|---|---|---|
| FAISS Baseline | 3265 | 0.707 | 2845 | 0.616 |
| DiskANN | 6503 | 2.876 | 17201 | 7.607 |
| CuANNS-IVFPQ | 108302 | 6.753 | 109745 | 6.843 |
| CuANNS-Multi | 839749 | 23.25 | 584293 | 16.18 |
| Ours | 46712 | 36.12 | 32608 | 25.22 |

---

[1]The benchmarked machine had eight GPUs, but only one was used, so we only account for one GPU's cost.

