# OpenReview forum: "SOAR: Improved Indexing for Approximate Nearest Neighbor Search"
_NeurIPS.cc/2023/Conference — NeurIPS 2023 poster_

### Official Review · Reviewer_jSJp · 2023-06-29

**Soundness:** 4 excellent
**Presentation:** 4 excellent
**Contribution:** 4 excellent
**Rating:** 8
**Confidence:** 3

**Summary:**

The paper proposes a neat variant of spill trees for approximate nearest neighbor search (ANN). The idea is to reduce correlation between the set of points retrieved from each of the partition elements in a spill tree (where the partitions elements overlap or "spill over") by adding a penalty term in the assignment function that involves the expectation of the angle between the query $q$ and the quantization residual $r$ (see Eq. (2)).

The idea is logical and the empirical results indicate state-of-the-art performance in large-scale benchmarks.

**Strengths:**

+ important problem
+ clear and sound idea
+ rigorous theory
+ outstanding empirical results
+ a pleasure to read

**Weaknesses:**

- the theoretical assumption underlying the derivation of the practical assignment formula (Thm. 3.1) assumes that the query distribution is uniform over a d-dimensional unit sphere, which is an unrealistic assumption -- however, the resulting method seems to perform well enough, so apparently the performance isn't sensitive to this

**Questions:**

detailed comments:

p. 2: "top k' partition": the use of the term partition to refer to the parts that make up the partitioned set is a bit problematic. I would prefer to use the term 'partition element' (or perhaps 'part' or 'cell') to distinguish the parts from the partition which includes them.

p. 4: is the axis label "quantized score error $\langle q,r \rangle$" (also mentioned in the heading of Sec. 3.2) a definition of a new term? if yes, it should be given in the main text

**Limitations:**

Yes, the authors have adequately addressed limitations.

---

> ### Author Rebuttal · Authors · 2023-08-09
>
> Thank you for your review. We’ll address the weaknesses and questions point-by-point:
> ### Weakness
> * We agree, this assumption isn’t entirely accurate. However, we’ve found this assumption to still be very effective, because it’s somewhat of a worst-case; the hardest queries tend to be those far from any datapoint, which a uniform query distribution produces. Real queries tend to have some correlation with the dataset distribution, but targeting improvement on the most difficult queries evidently seems to improve general performance as well.
> ### Questions
> * Just to make sure we understand this correctly, you are referring to distinguishing between the geometric concept of a partition (roughly, some sort of Voronoi cell), as opposed to the partitioned set of datapoints, and you prefer “partition” be used for the former idea? We can tweak that line to say “datapoints ***within*** the top k’ partitions” to emphasize the geometric nature of containment; would that help?
> * We will amend Section 3.1 to explicitly define quantized score error as $\langle q,r\rangle$ in the beginning of line 85; thanks for the suggestion.

---

> > ### Comment · Reviewer_jSJp · 2023-08-15
> > **On the term "partition"**
> >
> > * I was merely commenting on the use of the term 'partition' to refer to the "cells" or "partition elements" that make up what I'd call a "partition". For example for a set {1,2,3,4}, I would say that {{1,2}, {3,4}} is a partition composed of partition elements {1,2} and {3,4}. On lines 61-64 on p. 2, you seem to be using the term "partition" to refer to such partition elements, e.g., "We can then construct an inverted index over $\pi$; for each partition $i$, we store the set of data point indices belonging to that partition."

---

> > > ### Author Response · Authors · 2023-08-16
> > > **More on "partition"**
> > >
> > > Understood; we will re-think this phrasing. Our initial submission would refer to {{1,2}, {3,4}} as a “partitioning,” {1,2} as a “partition,” and 1 as a partition element; from our understanding, you are suggesting referring to {{1,2}, {3,4}} as a “partition,” {1,2} as a partition element, and using a new term for 1--this is perhaps more in line with mathematical norms, and we can adjust our terminology accordingly.

---

### Official Review · Reviewer_LtdJ · 2023-07-04

**Soundness:** 4 excellent
**Presentation:** 3 good
**Contribution:** 3 good
**Rating:** 7
**Confidence:** 5

**Summary:**

This paper introduces an innovative technique for constructing a vector index to perform approximate nearest neighbor search, building upon the foundations of vector quantization and redundancy through spill trees. The authors propose a novel loss function aimed at identifying k-alternate centroids in the vector quantization process, thereby minimizing the failure correlation between assignments to the original and alternate centroids. This reduction in correlation effectively decreases the likelihood of missing a nearest neighbor during the search phase. Notably, the authors substantiate their approach by showcasing improved recall@k scores on benchmark datasets, all achieved with minimal computational overhead.

**Strengths:**

* The ideas introduced in the paper are about spilling and redudancy in cluster assignments which have been previous explored in different capacities in the past. But the paper gives meaningful improvements on this task of approximate nearest neighbor(ANN) search which is a critical real world application.
* The authors' approach is characterized by its simplicity and ease of implementation, supported by a solid theoretical foundation. The motivation behind the proposed loss function is well articulated, emphasizing the necessity of reducing the correlation between <q,r> and <q,r'> while promoting orthogonality among the residuals.
* The paper is well-written and effectively conveys its ideas, providing adequate background information for readers to comprehend the research context.
* The recall@100 results achieved by the proposed method are notably strong, particularly on larger-scale datasets, underscoring its efficacy in handling real-world scenarios.
* The benchmarks conducted demonstrate satisfactory throughput results, indicating that the approach imposes minimal memory and computation overhead. This attribute enhances the practical viability of the method for deployment in real-world applications.
* The effectiveness of the proposed method scales well with increasing dataset sizes, as evidenced by the more pronounced improvements observed when dealing with larger datasets where the probability of failure in cluster assignments tends to rise.
* The authors further enhance the credibility of their findings by providing a compelling correlation analysis, elucidating the reasons behind the success of their proposed method (referred to as SOAR).

Overall, the paper presents a valuable contribution to the field, offering practical enhancements to approximate nearest neighbor search while maintaining simplicity in implementation. The results obtained through rigorous experimentation and analysis demonstrate the effectiveness and scalability of the proposed method.

**Weaknesses:**

* Section 3.5 lacks a lot of details and paper could heavily benefit from providing a more structured and detailed explanation of the final algorithm. For eg. clarifying whether the spilling is done hierarchically and addressing corresponding memory considerations.
* The improvement over previous SOTA ScaNN is marginal, and it would be helpful to if authors could elaborate more on the trade-offs and benefits over ScaNN
* The paper would also benefit from including more details in Section 4.1 about the comparison with spill trees and referencing relevant literature, making it more comprehensive and easier to read.
* A formal theoretical analysis of the time and space complexity would be valuable, providing a deeper understanding of the computational requirements of the proposed algorithm.

**Questions:**

* It would be useful to know if only one alternate centroid is used for each data point or if multiple alternate centroids are employed. If multiple centroids are utilized, it would be informative to see the effect of increasing the number of alternate centroids on the recall performance.
* In Figure 8, it would be beneficial if the authors provided a proper definition of "distortion" and explained why it matters. Additionally, offering more details on how to effectively choose the parameter λ would enhance the comprehension of the research findings.
* Suggestion: Authors can highlight the fact that decreased correlation between r and r’ means probability of misses decreases exponentially as you add more and more redundancy, through some formal analysis?
* Suggestion: The community would benefit from incorporating soar into common open-source tools like FAISS and ScaNN and increase it’s adoption

**Limitations:**

Yes the authors have adequately addressed limitations

---

> ### Author Rebuttal · Authors · 2023-08-09
>
> Thank you for your review. We’ll address the weaknesses and questions point-by-point:
> ### Weaknesses
> 1. We’ve added Figure 2 in the rebuttal PDF to better describe the implementation details and space complexity. We also plan on adding algorithm blocks to the appendix to more fully describe our indexing and query-time changes. At a high level, though, the indexing change is simply using our new loss to pick a second centroid assignment for each datapoint, and the only querying change is deduplicating results. We also already provide indexing source code, and plan on upstreaming our changes to ScaNN.
> 1. We see SOAR as not an alternative to ScaNN, but rather an add-on. For example, we continue to use the anisotropic vector quantization introduced by ScaNN, but additionally add our SOAR-loss spilled assignment. The primary tradeoff, as described in Table 1, is memory consumption; in Glove, we use +7.7% (submitted version says +12.7%, but this was measured incorrectly) additional memory, but get better performance. On Glove, the gain is small, but it’s also a small dataset, and as explained in Section 5.3, SOAR increases in efficacy as dataset size grows. That’s in part why the SOAR/no-SOAR gain is so much larger in Figure 10b.
> 1. We strived to include as much detail as possible while obeying the page limit; are there specific points in the comparison that you found confusing, or important characteristics of spill trees from literature that you felt should be included? While we likely won’t have the space to expand significantly upon this section, we could re-prioritize its exposition to focus on details you may believe to be more important.
> 1. Addressed in #1.
> ### Questions
> 1. Just one alternate centroid is used. Due to the linearly increasing storage cost of further assignments, but the diminishing benefits in terms of search efficiency gain (as measured by KMR; further alternate centroids by definition will have greater $||r’||^2$ or greater correlation, and will naturally be inferior), we found one alternate centroid strikes the best tradeoff. Also see the end of our rebuttal for Reviewer ixrh.
> 1. We implicitly define distortion in Figure 8’s caption: “Raising SOAR $\lambda$ increases VQ distortion $E[‖r’‖^2]$.” This matters because the naive way of choosing a spilled assignment would be to minimize $||r’||^2$. As demonstrated in Theorem 3.1, our loss includes the $||r’||^2$ term, but also an additional “orthogonality” term, and Figure 8 makes it clear how $\lambda$ adjusts the tradeoff between the two.
>    * To emphasize the $||r’||^2$ meaning, we’ve amended the third paragraph of Section 3.3 to now read: “This approach leads to some implementation intricacies, as discussed in Section 3.5, but for now we focus on problems concerning theory when the spilled assignment $\pi′(x)$ is chosen so as to minimize $||r′||^2$.
> 1. This would be a nice result, but we see it as a non-trivial and separate research endeavor, primarily because proving results regarding order statistics (which the top-k neighbors fundamentally boils down to) is difficult; even if $\langle q,r\rangle=0$ for a nearest neighbor, other non-nearest neighbors could have negative $\langle q,r\rangle$ that result in even higher approximate inner products, so the nearest neighbor may still be missed. The dependence of each vector on every other vector’s quantization residuals makes analysis challenging.
> 1. We have already released a fair amount of source code in the supplementary materials that can reproduce our results and shed light on implementation. Our results in Figure 10a came from patching the ScaNN source code and we're quite confident we can integrate those patches into the library, too.

---

> > ### Comment · Reviewer_LtdJ · 2023-08-16
> >
> > Thank you, the authors have addressed my questions satisfactorily. I would like keep the rating which is already accept.

---

> > > ### Author Response · Authors · 2023-08-16
> > > **Acknowledged**
> > >
> > > Thank you for taking the time to carefully review our work.

---

### Official Review · Reviewer_ixrh · 2023-07-06

**Soundness:** 3 good
**Presentation:** 3 good
**Contribution:** 2 fair
**Rating:** 4
**Confidence:** 3

**Summary:**

The paper discusses how a data index is built for Approximate Nearest Neighbor Search (ANNS). However, existing methods train and compute redundant representations independently, leading to inefficiency and inaccuracy. The paper proposes a novel data indexing technique called SOAR for approximate nearest neighbor (ANN) search, which utilizes multiple redundant representations and an orthogonality-amplified residual loss, drastically improving the overall index quality, resulting in state-of-the-art ANN benchmark performance.

**Strengths:**

1. The paper studies a very important and a very fundamental problem that finds a lot of applications in search tasks.
2. This paper has a clear motivation, i.e., increasing search efficiency when <q,r> is high. This idea is technically sound.
3. Many analyses and experiments have been conducted.

**Weaknesses:**

1. Both Preliminary and Method are based on the MIPS problem. However, the billion datasets of the experiment are not focused on the inner product completely (Microsoft SPACEV and Microsoft Turing-ANNS are L2 datasets).
2. Results of recall-QPS curves on more datasets (such as Yandex Text-to-Image, whose metric is the inner product) are needed.
3. Lack of discussion on the complexity of storage under spilled VQ assignment. Statistics in Tab. 1 are not detailed enough.

**Questions:**

1. Why use the L2 datasets to study the efficiency of MIPS methods? maybe more details on datasets are needed (How to use datasets? Which metric are datasets based on? )
2. Include more results on different datasets which are based on inner product metrics (such as Yandex Text-to-Image) or explain the reasonability of existing datasets.
3. Discuss the space cost of spilled VQ assignment with relative notations.
4. Please add more details about the loss function. How to generalize the loss function if the spilled assignments are above 2 ? (mentioned on line 126: This idea can be extended to further (>2) assignments as well.)

**Limitations:**

Please refer to Paper Weakness.

---

> ### Author Rebuttal · Authors · 2023-08-09
>
> Thank you for your review. W1/W2 + Q1/Q2 are all with regards to datasets chosen for experiments, so we will address this first.
> * Our initial paper involved experiments on four datasets, and we believe this is more than sufficient to prove our point. L2 and angular datasets are simply more common and widely benchmarked, with more points of comparison; we didn’t include Yandex Text-to-Image initially because it didn’t have results from the CuANNS-Multi algorithm, and we wanted to maximize our number of points of comparison.
> * We have now added Yandex Text-to-Image to our results as requested, and they’re shown on Figure 1 of the rebuttal. We achieve top performance there, too.
>
> W3/Q3: Figure 2 from the rebuttal should address your concerns about space complexity analysis. Also see Section 3.5 of the paper.
>
> Q4: Although we don’t have the space to explain in this rebuttal, the loss can be generalized by summing projective penalties to all previous assignments, when there’s more than two assignments. This generalizes the correlative penalty expounded in Section 3.4. This is generally not worth doing due to linearly increasing storage cost, compared to the diminishing returns in the positive effects of further assignments. >2 assignments is not used in any experiment.

---

### Official Review · Reviewer_J7HP · 2023-07-08

**Soundness:** 2 fair
**Presentation:** 3 good
**Contribution:** 2 fair
**Rating:** 4
**Confidence:** 3

**Summary:**

The paper revists the "vector quantization" technique for nearest neighbor search. In the VQ technique, points are represented by their closest centroid in a partition schema, typically based on k-means. Answering a query consists of finding the nearest centroids to the query and enumerating points represented by each centroid in the nearest centroid list.

Given that the nearest neighbor of a query need not be in the centroid closest to the query, one typicall has to explore a few centroids to find the right answer. In this paper, the authors study alternate schemes to reduce the number of centroids to explore.

Their contention is that centroids should be designed to minimize <q,r> where r is the residual x-C(x), of x, the nearest neighbor of q and its centroid C(x). They further focus on cos \theta, where \theta is the angle between q and r. Apart from the first eucliden residual minimizing partition scheme, the authors design a second cos \theta minimizing partition scheme.

collectively the intention is to reduce the depth of exploration.

The results are constrasted with existing techniques FAISS and DiskANN and on benchmark datasets.

**Strengths:**

A clear explanation of the techniques involved and visualization of why alternate spill over schemes dont work.

Empirical measurements indicated the rank of target vectors is improved by new design (Fig. 5.1)

Experiments comparing to ann-benchmarks and also cost-normalized metric on big-ann-benchmark. Strong results overall.

**Weaknesses:**

The index design  5.4 is unclear. line 276-278 need a bit more explanation -- what is the exact index and query path design.

The results in Fig 10b needs more explanation and information to be verifiable. There is no plot of QPS alone. Nor a report of the parameters used for baselines for others to judge if they are the best config. For DiskANN, I suspect the IO/s of a GCP local SSD (680K for 3.2TB ssd: https://cloud.google.com/compute/docs/disks/local-ssd) is not enough for 72 cores. Performance might saturate below what 72 cores issuing queries might need (in which case DiskANN must be priced in table A.3 with fewer cores).

**Questions:**

1. Can you please provide more details of the index and query algorithms

2. Can you please provide more details about how your arrived at plot 10b and what care was taken to prepare baselines.

3. How does this idea compare this idea with ScaNN which optimizes for angular metrics too, on large datasets.

4. What is the performance on other (non-Glove)datasets on ann-benchmarks.com

5. Can your ideas also be applied to product quantization (in each chunk) and improve PQ's error (normalized for total bytes)?

---

> ### Author Rebuttal · Authors · 2023-08-09
>
> Thank you for your review. To answer your questions point-by-point:
> 1. We’ve added Figure 2 in the rebuttal PDF to better describe the implementation details and space complexity. Also see Section 3.5 and Appendix A.3 for details. We also plan on adding algorithm blocks to the appendix to more fully describe our indexing and query-time changes. At a high level, though, the indexing change is simply using our new loss to pick a second centroid assignment for each datapoint, and the only querying change is deduplicating results. We also already provide indexing source code, and plan on upstreaming our changes to ScaNN.
> 1. To summarize our response to your concerns about Figure 10b, our baseline points of comparison were taken directly from the official big-ann-benchmarks site, so they (including their configs) are exactly explained and verified by the official procedure. To provide some more detail:
>    * The QPS numbers were taken directly from [here](https://github.com/harsha-simhadri/big-ann-benchmarks/blob/8a2947a3c1dca5c9f41bac4cde71a001698e44cc/neurips21/t3/LEADERBOARDS_PUBLIC.md#msspace-throughput-rankings) and [here](https://github.com/harsha-simhadri/big-ann-benchmarks/blob/8a2947a3c1dca5c9f41bac4cde71a001698e44cc/neurips21/t3/LEADERBOARDS_PUBLIC.md#msturing-throughput-rankings). Table 1 in the rebuttal includes a copy of that data so that our throughput/cost ratio calculations can be more easily verified.
>    * Regarding DiskANN performance: we took hardware details directly from the big-ann-benchmarks submission, and the DiskANN submission chose to use a setup with 2x18=36 physical cores (=72vCPU); see [here](https://github.com/harsha-simhadri/big-ann-benchmarks/blob/f146969aab6e524b7a5a11900d9eaa91c9c412d3/t3/diskann-bare-metal/README.md). Given that the submission author is a lead DiskANN contributor, we assume this hardware selection was an intelligent choice that caters to the strengths of the algorithm. Had less CPU power been sufficient, they could’ve submitted with less CPU and achieved better rankings on the power and cost leaderboards.
> 1. We see our work as an extension of ScaNN (we continue to use the anisotropic vector quantization technique introduced by ScaNN, and add SOAR on top), and have comparisons to vanilla ScaNN as a baseline. ScaNN without SOAR is shown in Figure 10a and a no-SOAR comparison is also present in Figure 10b.
> 1. With the scarce amount of extra space given in the rebuttal, we’ve decided to demonstrate performance on Yandex Text-to-Image rather than an ann-benchmarks dataset. This was because the Yandex dataset is more novel (unnormalized MIPS distance, not angular distance), and larger scale, where ANN research is most important (ann-benchmarks size datasets are easily solved with accelerators like TPUs at extremely high throughput and low cost). Furthermore, our supplementary materials include source code that allows anyone to easily perform explorations on additional ann-benchmarks datasets if they so wish.
> 1. We do not believe this is applicable to PQ. In PQ, one can (for example) double the bitrate by halving the size of each chunk, or squaring the number of centers per chunk. In VQ, neither of the above works (there are no such things as chunks; squaring, for example, 1M VQ centers would lead to 1T centers, which is totally impractical because that’s far greater than the number of datapoints) so the spilling idea is needed.

---

> > ### Comment · Reviewer_J7HP · 2023-08-15
> > **Cost calculations are off**
> >
> > For 10b, please consistently quote QPS and price from one source.
> > For example, DiskANN is measured at 17000QPS for MSTuring in the table you cite, and the capital cost is 11000$ or less than $300/month. You instead substitute a price of $2200/month based on your cloud costs. In track 3 of big-ann-benchmarks you cite, participants were given asked to jointly optimize hardware and algo configuration. So, if you were going to use cloud set up cost, then it is not correct to cite algo/hardware configure reported for a totally different set up. Find the optimal setup for baselines on GCP and then report the numbers. In my review I suggested that given cloud SSDs have lower IO/s rating than bare metal -- you should correspondingly use fewer cores for the cost-optimal setup, otherwise you would be wasting cores blocked on IO.
> >
> > On Optane, the winning entry on T3, you could easily substitute optane for memory and report the results for HNSW or DiskANN (in-mem) -- they would be quite compelling too.

---

> > > ### Author Response · Authors · 2023-08-16
> > > **Response to costs**
> > >
> > > Thank you for your continued interest in our work. We believe there may remain some misunderstandings regarding Figure 10b and are keen to dispel any beliefs you may have about unfair comparison or improper procedure. Here are two homologous responses to your concerns:
> > >
> > > 1. From your comments, it seems you find QPS/initial capital to be a fair comparison, because those numbers can be directly derived from big-ann-benchmarks. So here are those ratios in a table (for MS-Turing):
> > >
> > >    **Algorithm**|**QPS**|**Cost**|**Ratio**
> > >    :-----:|:-----:|:-----:|:-----:
> > >    FAISS Baseline|2845|22021.90|0.1291895795
> > >    Gemini|21780|55726.66|0.3908362712
> > >    DiskANN|17201|11742.37|1.464866122
> > >    CuANNS-IVFPQ|109745|150000|0.7316333333
> > >    CuANNS-Multi|584293|150000|3.895286667
> > >    Ours|32608|6814.04|**4.785413646**
> > >
> > >    The \\$150K figure for CUANNS was the official estimate the big-ann-benchmarks organizers used. We can also provide numbers for the other datasets; we’re still first in those, too. Our \\$6814 figure comes from a PowerEdge R740 with a Xeon 6208U and 5x32GB RAM. If you follow the link from the DiskANN big-ann-benchmarks submission’s README.md to the Dell website, and configure as we have, you should get an identical price (links are not allowed in comments). While we unsurprisingly didn't have the time to purchase, set up, and benchmark a PowerEdge R740 in the past day, we would be very surprised if it couldn't achieve 32608 QPS on such a machine, given our GCP instance had a significantly lower clock rate and also had to contend for shared resources (like memory bandwidth) with other jobs on the shared cloud infrastructure.
> > >
> > > 1. \\$2200/month is in fact a very reasonable monthly cost for hosting the benchmarked DiskANN setup on a cloud provider. You would find a similar figure if looking on AWS, Azure, or any other major cloud provider. We're unsure of how you derived \\$300/month from a capex of \\$11000, but perhaps you are forgetting energy costs, administrative expenses, or the need for these providers to maintain a profit margin. At any rate, even if this figure is correct, using that same capex/monthly cost ratio would imply our technique would cost \\$185/month ($300\times6814/11000$), and the relative ranking would remain unchanged. Regarding cloud SSD IOPS, this would mean we’re generously underestimating DiskANN’s cost, because if your claim were true, that would imply we’d have to spend more on cloud SSD to effectively utilize the 72vCPU and achieve the same QPS.
> > >
> > > Ultimately, we are confident that our experimental results are broadly competitive under reasonable evaluation criteria.

---

> > > > ### Comment · Reviewer_J7HP · 2023-08-21
> > > >
> > > > I am not denying your own measurements, which are good, as much as I am suggesting that your reporting of baselines is not complete and accurate.
> > > >
> > > > 1. If you want to cite big-ann-benchmarks, please cite all results. Especially the top entry -- you can't leave that out. If your argument is that Optane pmem is not publicly available now, so is the case with Gemini. One could also argue that replacing pmem with DRAM could give comparable, if slightly lower, numbers for Intel entries (and it does, BTW).
> > > > 2. To your point on "underestimating DiskANN’s cost": As I mentioned above, if you are trying to optimize QPS/cost, you would use fewer cores for the same SSD and get the similar QPS, and improve QPS/cost since cores are the most expensive component on cloud hardware. My point is that for each algorithm, you have to find the best possible configuration for the setting you are reporting (in this case GCP), rather than picking hardware configurations chosen in another setting (in this case Dell PowerEdge with a local SSD with higher IO/s).
> > > > 3. Re. 300 USD cost, the competition rules don't ask for administrative expenses AFAIK. You can divide the hardware capital cost by 48 months and add (~1kW * 720 hours * 10cents/kWh) in power per month. The number is $315. The point is not that you should buy a Dell and experiment. It is that every entry is allowed to pick algo and hardware and self-report.
> > > > 4. In the cost and QPS table in your response, how is the hardware cost the same for both CuANNS entries -- $150K for both single and dual multiple GPUs? Shouldn't the hardware cost for CuANNS-IVFPQ be lower than for multi-GPU setting?

---

### Author Rebuttal · Authors · 2023-08-09

Thank you all for taking the time to review our work. The two most common points of contention and confusion among the reviews were with regards to our experiments and with implementation details (especially memory footprint). We believe we have addressed both in our rebuttal. Our added material for the rebuttal consists of three components:
* The first describes exactly how our points of comparison for Figure 10b of the experiment in Section 5.4 were computed.
* The second adds a new dataset, Yandex Text-to-Image, an unnormalized billion-vector MIPS dataset from big-ann-benchmarks.
* The third describes the memory layout for a SOAR-enabled index and the memory overhead incurred by SOAR.

We naturally plan on including the above figures in the final version of our paper, as well as additional expository text which, due to the rebuttal rules, we were unallowed to showcase here. Individual critiques are addressed per-review; please feel free to ask additional questions or clarifications.

---

### Decision · Program_Chairs · 2023-09-21

**Decision:**

Accept (poster)

**Comment:**

Two reviewers recommend clear Accept (one strong accept) citing that the paper has a clear motivation, i.e., increasing search efficiency when the distance of the query to the residual of the closest centroid is high, and propose a novel solution.

The ACs agree that the paper presents a novel approach and recommend acceptance. They however also agree with Reviewer J7HP who concerned about the main comparissons of the paper in Figure 10b. Fir the final version, the authors are strongly urged to:

* Clearly and extensively explain the experimental setup for 10b: a complete account (details can be in supplementary) on the experimental setup for all methods, including cost calculations and any new info from the discussions during rebuttal.

* Cite more top results from big-ann-benchmarks, including top entries that might not publicly available. The ACs agree that such results should be in the table, with proper notation when there are differences in setup or where the numbers are sourced from (if models not publicly available).